# Simple ETₒ-Based Rules for Irrigation Scheduling by Smallholder Vegetable Farmers in Laos and Cambodia

John McPhee [1,*], Jochen Eberhard [2], Alice Melland [2], Jasim Uddin [3], Lucinda Dunn [4], Sarith Hin [5], Vanndy Lim [5], Veasna Touch [5], Phimmasone Sisouvanh [6], Inthong Somphou [6], Tounglien Vilayphone [7], Phaythoune Mounsena [7] and Stephen Ives [8]

1   Tasmanian Institute of Agriculture, University of Tasmania, Private Bag 3523, Burnie, TAS 7320, Australia
2   Centre for Agricultural Engineering, University of Southern Queensland, West St., Toowoomba, QLD 4350, Australia; Jochen.Eberhard@usq.edu.au (J.E.); Alice.Melland@usq.edu.au (A.M.)
3   Soil and Water R&D Unit, NSW Department of Primary Industries, Trangie Agricultural Research Centre, Trangie, NSW 2823, Australia; jasim.uddin@dpi.nsw.gov.au
4   School of Life and Environmental Science, University of Sydney, Sydney, NSW 2006, Australia; lucinda.dunn@hotmail.com
5   Cambodian Agricultural Research and Development Institute, National Road No 3, Preteah Lang Commune, Dangkor 12413, Cambodia; sarith.hin@gmail.com (S.H.); vanndylim168@gmail.com (V.L.); veasna80@gmail.com (V.T.)
6   Faculty of Agriculture, National University of Laos, Nabong Campus, Vientiane 0106, Laos; phimmasone2004@yahoo.com.au (P.S.); inthongsomphou@yahoo.com (I.S.)
7   Horticultural Research Centre, National Agriculture and Forestry Research Institute, Vientiane 0106, Laos; tounglien@gmail.com (T.V.); rattanaphaythoune@yahoo.com (P.M.)
8   College of Science and Engineering, University of Tasmania, Locked Bag 1354, Launceston, TAS 7250, Australia; stephen.ives@utas.edu.au
*   Correspondence: john.mcphee@utas.edu.au

**Abstract:** Hand-held hoses and watering cans are widely used by smallholder farmers to irrigate vegetables in Cambodia and Laos. Overwatering is common. Technology change (e.g., low-pressure drip irrigation) has been used to improve irrigation efficiency but can be unaffordable for many smallholder farmers. The purpose of this study was to identify an appropriate method of predicting crop water demand, develop and field-test improved irrigation schedules for smallholder leafy vegetable farming based on that method, and then develop extension tools to communicate the schedules to smallholder farmers. Improved irrigation schedules for leafy vegetables were developed based on a crop water use prediction technique that is well established (the Penman–Monteith method) but beyond the capacity of smallholder farmers to implement without access to simple aids. Compared to conventional practice, the method approximately halved water and labour use and improved irrigation water productivity 2–3 fold in field research and demonstration trials. Simplified extension tools to assist smallholder farmers with practice change were developed. This work showed that significant efficiencies could be gained through improved irrigation scheduling without changing application technology.

**Keywords:** irrigation; leafy vegetables; evapotranspiration; nomograph; water productivity

## 1. Introduction

Domestic vegetable production is promoted by the governments of both Laos and Cambodia to improve food security [1,2] as both countries are net importers of fresh vegetables. Most domestically produced vegetables are grown by smallholder farmers [3,4]. Smallholder farm incomes can be improved, and rural poverty reduced, through production of horticultural crops if practice change is not constrained by factors such as affordability, input supplies, technical support, market access, land use policy, and post-harvest technology [5,6].

Laos and Cambodia have strong seasonal rainfall patterns, with a monsoon-driven wet season (May–October) and a dry season (November–April). Temperatures are high throughout the year except in elevated areas, which comprise a relatively small area of the two countries. Most agricultural production (other than rice) occurs in the dry season, necessitating irrigation.

Watering by hand-held hose or watering can is common amongst smallholder vegetable farmers and is time-consuming and labour intensive [3]. Low-cost drip irrigation is one alternative to hand watering for smallholder vegetable farms [3–5]. Despite potential advantages of reduced labour and water use, fewer weeds, and improved yield, adoption of such technology may be limited by the capital cost, lack of component availability, and limited support for system design and operational training [4].

However, changes in technology are not the only aspect of improving irrigation performance. While drip irrigation has many potential benefits, unrealistic expectations of system performance, in terms of area watered, and the use of non-regulated dripper systems, may lead to suboptimal application rates and poor distribution uniformity (DU) [7]. This underscores the importance of understanding crop water demand and the need for scheduling irrigation to improve irrigation management, regardless of the technology used.

Irrigation scheduling, which determines the frequency and volume of water applied to a crop [8], is essential to avoid soil water excess or deficiency, both of which can reduce yield, crop quality, and water use efficiency (WUE), and increase plant disease risk and soil degradation [9–13]. Over-irrigation can also increase water and nutrient loss in runoff and drainage and increase pumping energy costs [12]. Effective irrigation management is a complex decision-making and field application process [8]. Scheduling involves knowledge of crop water requirements, yield responses to water, security of water supply, weather influences, soil infiltration characteristics, field soil condition, application uniformity, performance of different types of irrigation systems (e.g., pump and pipe capacity to supply the required amount, sprinkler output etc.), and the financial implications of irrigation practice. Due to this complexity, irrigation scheduling is not widely used [8] and instead, it is common for farmers to irrigate based on past experience [12,14]. Irrigation scheduling techniques have been found to improve yield and WUE in vegetables [10,11]. However, there has been limited irrigation scheduling research specifically for vegetable production in Cambodia or Laos.

There is a range of irrigation scheduling methods, including those based on weather (e.g., FAO Penman–Monteith, evaporation pans) [9,10,12,15], soil moisture sensors (e.g., volumetric, soil moisture tension) [12,16,17] and plant-based sensing [12,18]. All these methods require some training and skills for their effective use, and the cost of sensor-based systems is often beyond the capacity of smallholder farmers in developing countries [19]. Simple, low-cost irrigation scheduling methods suited to hand watering are needed for such farmers to improve water and vegetable productivity. Tools suited to this task should have low capital, operational, and maintenance costs, be practical and simple to use, and provide an adequate return on investment to encourage adoption [4,5]. Other important factors to consider are the impact of irrigation scheduling on vegetable production and the effect of climate on water supply and choice of scheduling methods.

The Penman–Monteith method of estimating crop evapotranspiration from weather data can be an effective way of predicting irrigation requirements [15]. However, local weather data are limited in developing countries, making it necessary to rely on online open-source weather data (e.g., NASA POWER—https://power.larc.nasa.gov/data-access-viewer/) (accessed on 4 June 2022). The NASA POWER Agroclimatology data base contains over 35 years of records for all the data required (i.e., solar radiation, maximum, minimum and dew-point temperature, wind speed, and relative humidity) for the Penman–Monteith FAO 56 method of estimating crop water demand. The Penman–Monteith method has been widely used for scientific research over many years [20–25]. However, very few have attempted to present the workings of the Penman-Monteith model in a form that is useful

to farmers to assist with irrigation scheduling [26–28]. The mathematical processes required to estimate crop water demand are beyond the capabilities of smallholder farmers.

While other methods of estimating crop water demand were considered [19], the objectives of this particular study were to test the effectiveness of the Penman–Monteith method of estimating crop water use to inform irrigation scheduling for smallholder vegetable producers, and to refine the method to support practice change. To meet the objectives, the method was evaluated in field trials in smallholder vegetable farm settings, and a simple aid to facilitate the use of the irrigation scheduling method by smallholder vegetable farmers was developed.

## 2. Materials and Methods

### 2.1. NASA POWER Weather Data and $ET_o$-Based Method

NASA POWER Agroclimatology data were compared with weather station (Campbell Scientific, CR800 series) data recorded at the Cambodian Agricultural Research and Development Institute (CARDI), Phnom Penh, Cambodia (11.4766° N, 104.8079° E), during the period 22 March 2014 to 12 July 2017. Daily weather data recorded at CARDI included solar radiation, mean temperature, relative humidity, wind speed, and atmospheric pressure. Due to equipment malfunction at various times, the data were not continuous over the 1209-day period and covered a total of 834 days across three blocks of time that ranged from three to nineteen months' duration. NASA POWER weather data for matching periods were downloaded from the NASA POWER database. Individual day data were averaged to provide day of year (DOY) data. The daily reference evapotranspiration ($ET_o$) was estimated using the procedures in the Penman-Monteith model [15]. Pair-wise comparisons of each weather parameter were made using linear regression. Root mean squared error (RMSE), mean absolute error (MAE), percentage error of estimate (% Err of Est.), and standard error (s.e.) were used to evaluate the accuracy of NASA POWER predicted weather data compared to the weather station recorded data. Data were analysed, and predicted $ET_o$ was calculated using Microsoft Excel®.

### 2.2. Smallholder Farming Systems and Study Sites

Irrigation scheduling methods were tested at six field sites (Table 1). The field sites in Cambodia were located on the low-lying plains near Tonlé Sap Lake (near Siem Reap, ~20 m ASL), close to the southern coast (near Kampot, ~10 m ASL), and between the Bassak River and Dâmrei Mountains (near Chhouk, ~30 m ASL). The field sites in Laos were located approximately 30 km north-east and 12 km south of the capital Vientiane (both ~170 m ASL), with the site south of Vientiane being on the Mekong River flood plain.

Leafy salad and cooking vegetables (e.g., *Lactuca sativa*, *Ipomoea aquatica*, *Brassica rapa* subsp. *Chinensis*) comprise a major portion of the production of smallholder farmers in the regions that were the focus of this study. Standard farmer practice is to grow seedlings in a nursery bed until they are ready to be transplanted as bare-rooted seedlings to larger production beds. Transplant shock is evident after transplanting, and this is addressed by frequent watering (at least twice per day) and sometimes shading with available materials until the plants recover.

The dominant forms of irrigation used in the study areas were hand-held hoses with a shower spray head attachment or watering cans of either 12 or 15 L capacity. Overhead sprinklers and drip systems were uncommon. Irrigation water was sourced from captured rainfall, groundwater, and streams, and where required, stored in ponds and tanks near the cropping area. Typical irrigation practice was to water twice per day, once in the morning and once in the evening. The volume of water applied is typically decided based on surface soil wetness and plants showing no signs of stress.

**Table 1.** Location, soil, and climate of irrigation trial field sites in Laos and Cambodia.

| Country | Province | District | Village | Lat | Lon | Soil type | Climate [a] | Avg Annual Rainfall (mm) [a] | Average Min/Max Temperature (°C) [a] |
|---|---|---|---|---|---|---|---|---|---|
| Laos | Vientiane Capital | Xaithany | Pakxapkao | 18.1400° N | 102.7753° E | sandy clay loam to 30 cm | Tropical savanna | 1989 | 21.8/29.9 |
| | | | Nabong | 18.1244° N | 102.7919° E | | | | |
| | | Hadxaifong | Huaha | 17.0867° N | 102.6067° E | silt loam over silty clay loam to 30 cm | Tropical savanna | 1989 | 21.8/29.9 |
| Cambodia | Kampot | Tuek Chhou | Koun Sat | 10.5967° N | 104.2783° E | sandy loam over loam to 60 cm | Tropical monsoon | 1807 | 24.6/29.5 |
| | | Chhouk | Prey Ben | 10.8356° N | 104.4553° E | sandy loam over loam to 60 cm | Tropical savanna | 1500 | 23.5/30.8 |
| | Siem Reap | Prasat Bakong | Ta Trav | 13.3353° N | 104.0289° E | sandy loam over loam to 60 cm | Tropical savanna | 1358 | 23.0/31.4 |

[a] https://en.climate-data.org/ (accessed on 4 June 2022).

### 2.3. Baseline Observations and Measurements

### 2.3.1. Field Observations

Field observations during the early phases of this study indicated that although irrigation often involved frequent, high intensity applications of water, it was common to find that only the surface of the soil was wet, with zones lower in the profile remaining dry. Surface moisture evaporates quickly in tropical climates, particularly in the dry season and in early crop growth stages when there is minimal ground cover. Further, as the frequency and amount applied remained largely unchanged through the growing season, the amount of water applied at later growth stages may have been insufficient to meet crop demand as predicted from crop growth analysis and $ET_o$ estimation.

### 2.3.2. On-Farm Recording of Irrigation Practices

Leafy vegetable crop irrigation practice was recorded by farmers using irrigation diaries on four farms in Chhouk district, Kampot, Cambodia, during March–April 2018. Farmers recorded the timing of irrigation and the number of watering cans applied to the crop. Applied amounts, calculated on the basis of the irrigation diary data, were then compared to predicted crop demand as calculated by the Penman–Monteith method [15].

### 2.4. Field Experimentation and Demonstration Trials

### 2.4.1. Laos Experimental Designs

Irrigation experimental trials were conducted in the growing seasons of 2018 and 2019 in the Xaithany district of Vientiane Capital Province, at the National University of Laos (NUoL, 18.1244° N, 102.7919° E). The objective of the trials was to assess the effectiveness of using predicted $ET_c$ as a means of scheduling irrigation applications for lettuce (*Lactuca sativa*). The effect of different irrigation frequency (daily and every second day) was also tested. In the 2018 experiment, three irrigation treatments were assessed in conjunction with three lime rates in a split-plot, four-replicate design. In 2019, three irrigation treatments were assessed alone in a randomised complete block design with four replications. In both seasons, the irrigation treatments were irrigation rates (9 mm/day) and frequency (twice per day) based on observed farmer practice (farmer practice, I1), and irrigation rates based on predicted $ET_c$, which was calculated from NASA Power data and FAO guidelines [15], and applied once per day (I2) or once every second day (I3), except for the period immediately after transplanting, which used twice per day irrigation regardless of the treatment (Table 2). Values of the crop factor, $k_c$, used to estimate $ET_c$ for treatments I2 and I3 of this trial were based on values for leafy vegetables [15]. A value of 0.7 was used for the initial growth stage ($k_{c(initial)}$) after the transplant shock period (5–6 days) and 1.05 thereafter ($k_{c\,(mid)}$). As leafy vegetables are harvested fresh and consume water at peak rates right up to harvest, the $k_{c(mid)}$ value was used through until harvest, rather than transitioning to a $k_{c(late)}$ value. Table 2 shows the amount of water applied in the different irrigation treatments for the two seasons of field trials. The amounts applied each year

differed based on refinement of the method and differences in transplanting dates and hence predicted $ET_c$. All treatments were irrigated twice per day for the first 5–6 days after transplanting to ensure the bare-rooted transplants overcame transplant shock.

**Table 2.** Irrigation treatments applied to lettuce crops grown over two seasons at National University of Laos, Nabong, Xaithany district, Vientiane Capital Province, Laos.

| DAT [a] (Dates) | Irrigation Treatment [b] | | |
|---|---|---|---|
| | I1 | I2 | I3 |
| **2018** | | | |
| DAT 1–6 (31/3–6/4/18) | 9 mm/day (4.5 mm morning and evening) | 6 mm/day (3 mm morning and evening) | 6 mm/day (3 mm morning and evening) |
| DAT 7–14 (7–13/4/18) | 9 mm/day (4.5 mm morning and evening) | 4 mm/day applied each morning | 8 mm applied every second morning |
| DAT 15–28 (14–27/4/18) | 9 mm/day (4.5 mm morning and evening) | 4.5 mm/day applied each morning | 9 mm applied every second morning |
| **2019** | | | |
| DAT 1–5 (16–20/2/19) | 9 mm/day (4.5 mm morning and evening) | 3.7 mm/day (1.85 mm morning and evening) | 3.7 mm/day (1.85 mm morning and evening) |
| DAT 6–12 (21–27/2/19) | 9 mm/day (4.5 mm morning and evening) | 2.6 mm/day applied each morning | 5.2 mm applied every second morning |
| DAT 13–22 (28/2–6/3/19) | 9 mm/day (4.5 mm morning and evening) | 3.4 mm/day applied each morning | 6.8 mm applied every second morning |
| DAT 23–29 (7–16/3/19) | 9 mm/day (4.5 mm morning and evening) | 4.3 mm/day applied each morning | 8.6 mm applied every second morning |

[a] days after transplanting; [b] I1—farmer practice; I2—predicted $ET_c$ once per day; I3—predicted $ET_c$ once every second day.

### 2.4.2. Cambodia Demonstration Trial Designs

During the 2019 growing season, two irrigation methods were demonstrated without replication of treatments on three farms in Chhouk district, Kampot Province, growing mustard greens (*Brassica integrifolia),* and two farms in Ta Trav village, Prasat Bakong district, Siem Reap Province, growing bok choy (*Brassica rapa* subsp. *Chinensis)*. The two methods were farmer irrigation practice (twice a day, early morning and evening) and once per day, mid-morning irrigation using volumes based on predicted crop water demand [15]. Values of $k_c$ used to predict ETc for the demonstration trials were based on values for leafy vegetables [15]. Following the transplant shock period of up to 6 days, $k_c$ values used were 0.7 ($k_{c(initial)}$) and 1.05 thereafter ($k_{c(mid)}$). Details of the irrigation treatments are given in Table 3.

**Table 3.** Irrigation scheduling methods used at on-farm demonstration sites in Kampot and Siem Reap Provinces, Cambodia, in the 2019 growing season.

| Irrigation Treatment | Details |
|---|---|
| Farmer Practice | Kampot: 2 times/day, 8–12 mm per day Siem Reap: 2 times/day, 10–14 mm per day |
| Predicted $ET_c$ | 1 time/day, 3–5 mm per day according to predicted demand based on crop type, crop growth stage and province |

### 2.4.3. Field Measurements and Data Analysis

Yield was measured as the fresh weight of harvested vegetable on the day of harvest. Irrigation diaries were used to record the application rates and timing in line with the treatment protocols. No effective rainfall was recorded throughout the trial periods. The irrigation water productivity ($WP_I$) [29] of each treatment was calculated as crop yield divided by the volume of irrigation water applied. Statistical analysis of yield and irrigation data from the replicated field trials was done by analysis of variance using SAS v9.4 (SAS Institute Inc., Cary, NC, USA).

## 3. Results

### 3.1. NASA POWER Weather Data and $ET_o$-Based Method

Comparisons between weather data measured at the CARDI weather station and data retrieved from the NASA POWER database are given in Supplementary Materials (Table S1), while statistical performance evaluations are presented in Table S2. The difference between average $ET_o$ calculated from the weather data and from NASA POWER data was about 13%, with RMSE = 0.8 mm day$^{-1}$ and $R^2$ = 0.98. The relationship between the two sources of $ET_o$ estimation is given in Figure 1.

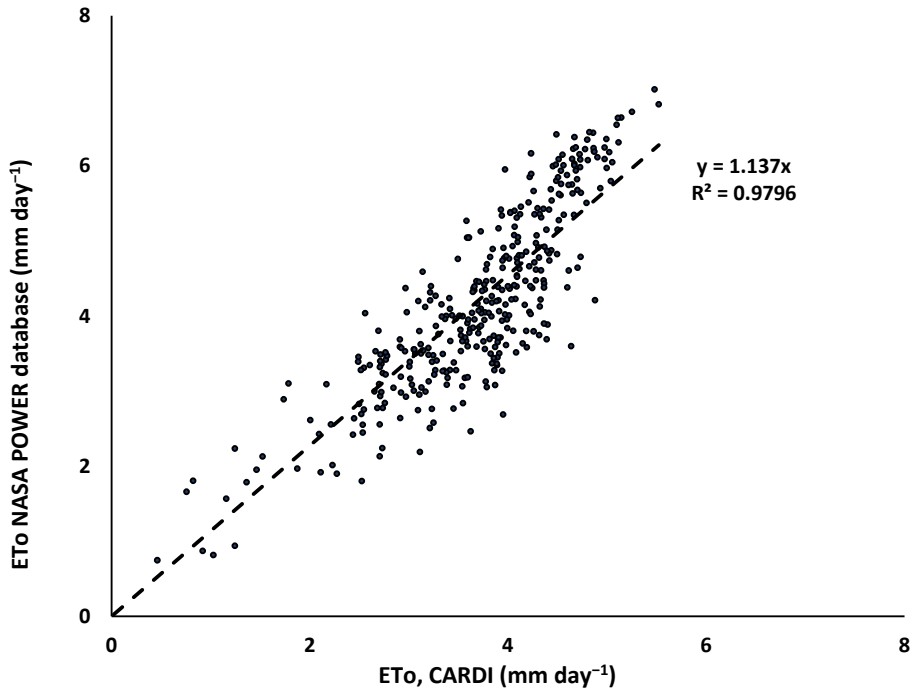

**Figure 1.** Relationship between $ET_o$ calculated from weather data measured at CARDI and estimated from corresponding NASA POWER weather data.

### 3.2. Baseline Measurements

On-Farm Recording of Irrigation Practices, Cambodia

Table 4 shows the total number of 15 litre watering cans, and the total depth of water applied to each of the crops monitored in Chhouk district, Kampot province, in 2018. The crops were all leafy vegetables with similar growth habits and growing periods (ranging from 21–24 days), so their water demand would have been similar. However, there was an almost 2-fold difference in the amount applied to the different crops across the four farms (Table 4).

**Table 4.** Total depth of water applied to similar crops in the same growing season on four farms in Chhouk, Kampot province, Cambodia.

| Farm No. | Total Number of Watering Cans | Total Depth of Water Applied (mm) | Average Depth of Water Applied (mm/day) |
|---|---|---|---|
| 1 | 134 | 168 | 8.0 |
| 2 | 166 | 208 | 9.4 |
| 3 | 92 | 115 | 5.0 |
| 4 | 96 | 120 | 5.0 |

The daily application at each farm compared to the average predicted daily $ET_c$ calculated using the Penman–Monteith method [15] from 33 years of NASA POWER data

is shown in Figure 2. The crops at farms 1 and 2 were over-watered for the whole growth cycle by almost double the required amount at various times. For most of the crop cycle, these two crops could have managed with just one watering in the morning. Within the realms of application accuracy with watering cans, irrigation at farms 3 and 4 was close to ideal, except for the 20% overwatering early in the growing season.

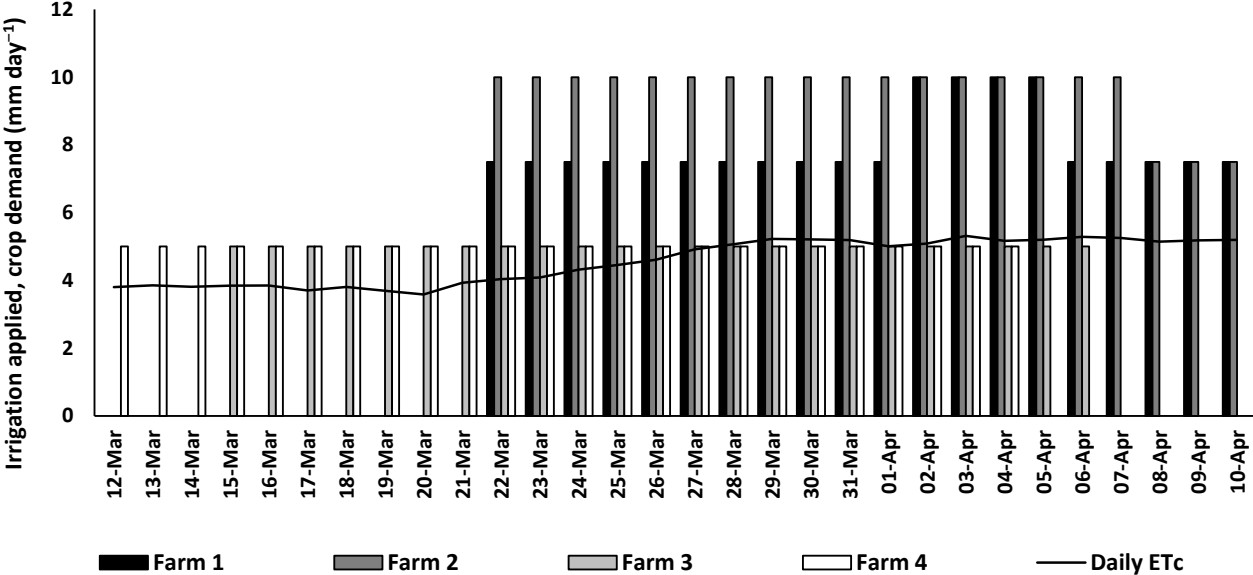

**Figure 2.** Daily application measured on four farms in Chhouk, Kampot Province, Cambodia, showing differences in the depth of water applied to similar crops in similar growing conditions, and compared to predicted $ET_c$.

### 3.3. Field Experiments and Monitoring

#### 3.3.1. Laos

Large differences in the amount of water applied were recorded between treatments, resulting in significantly higher irrigation water productivity in both years for the $ET_c$-guided irrigation treatments (I2 and I3) compared with farmer practice (I1) (Table 5). There were no statistically significant differences in yield between the irrigation treatments in either year of field experiments.

**Table 5.** Yield, water applied, and irrigation water productivity ($WP_I$) measured in two experiments at National University of Laos, Nabong, Xaithany district, Vientiane province, Laos. Different letters in the same column for each year signify statistically significant differences $p < 0.05$.

| Year | Treatment [A] | Yield (t ha$^{-1}$) Mean | s.e. | Water Applied (ML ha$^{-1}$) | $WP_I$ (kg ML$^{-1}$) Mean | s.e. |
|---|---|---|---|---|---|---|
| **2018** | I1 | 15.5 a | 1.41 | 2519 | 6.17 a | 0.97 |
|  | I2 | 16.6 a | 1.41 | 1330 | 12.50 b | 0.97 |
|  | I3 | 18.1 a | 1.41 | 1370 | 13.23 b | 0.97 |
| **2019** | I1 | 18.4 a | 3.82 | 2610 | 7.06 a | 2.49 |
|  | I2 | 22.8 a | 3.82 | 1035 | 21.89 b | 2.49 |
|  | I3 | 26.5 a | 3.82 | 1027 | 25.72 b | 2.49 |

[A] I1—farmer practice (water twice per day); I2—water once per day to predicted $ET_c$; I3—water once every second day to predicted $ET_c$.

### 3.3.2. Cambodia

Like the experiments in Xaithany district, Laos, crops in the trials in Cambodia irrigated using the $ET_c$-guided method of predicting crop demand used less than half the water of farmer practice without any noticeable difference in crop yield (Table 6). The result was a 2.6-fold improvement in $WP_I$ for both crops. The irrigation schedules for normal farmer practice, predicted $ET_c$, and the simplified $ET_c$ used for the field trials are shown in Figure 3.

**Table 6.** Yield, water applied, and irrigation water productivity ($WP_I$) for leafy vegetable crops in Kampot and Siem Reap Provinces, Cambodia, 2019, irrigated using farmer practice (twice per day) and with guidelines based on predicted $ET_c$ (once per day).

| Irrigation Method | Mustard Green | | | Bok Choy | | |
| --- | --- | --- | --- | --- | --- | --- |
| | Leaf Yield (kg ha$^{-1}$) | Water Applied (ML ha$^{-1}$) | $WP_I$ (kg ML$^{-1}$) | Leaf Yield (kg ha$^{-1}$) | Water Applied (ML ha$^{-1}$) | $WP_I$ (kg ML$^{-1}$) |
| Farmer Practice | 30,556 | 2107 | 14.5 | 24,400 | 2450 | 10.0 |
| Predicted $ET_c$ | 30,186 | 811 | 37.2 | 24,600 | 956 | 25.7 |

Data are the mean of three sites for mustard green and two sites for bok choy.

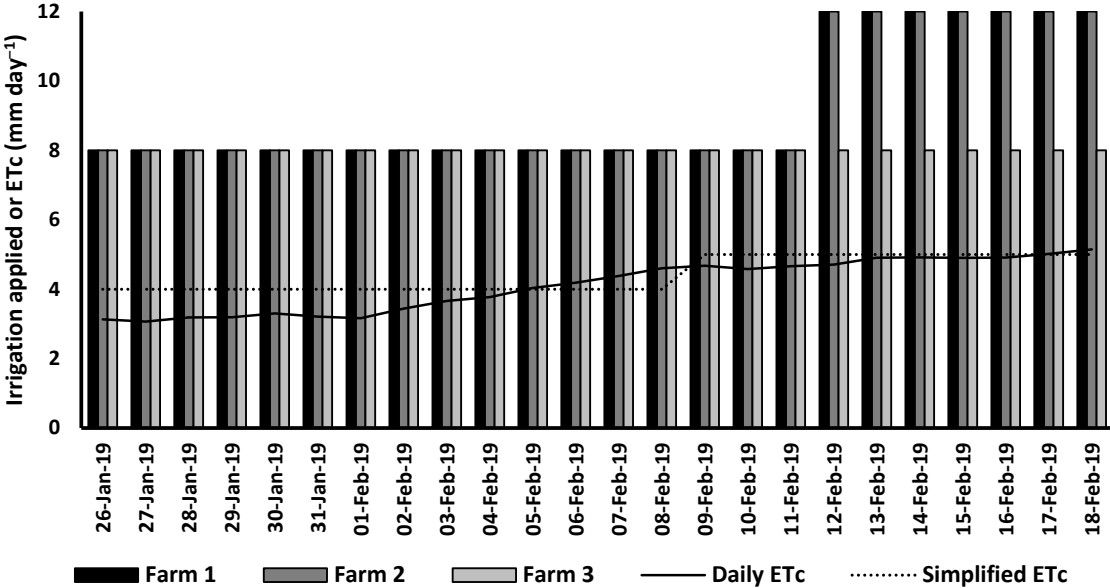

**Figure 3.** Daily application measured during demonstration trials on three farms in Chhouk, Kampot Province, Cambodia, showing differences in the amount of water applied to similar crops in similar growing conditions, and compared to predicted $ET_c$, and a simplified form of $ET_c$ used for demonstration of an alternative irrigation schedule.

### 3.4. Development of ETc-Based Irrigation Scheduling Approach Suited to Smallholder Farmers

Measurements of soil moisture and comparison of current irrigation practices against crop water demand as predicted by $ET_c$ (Penman–Monteith method) [15] showed that normal farmer irrigation practices in the study regions of Laos and Cambodia were suboptimal for the production of leafy vegetables. The results of replicated experimental trials in Xaithany district, Vientiane Capital Province, Laos, and farmer demonstration trials in Tuek Chhou and Chhouk districts in Kampot and Prasat Bakong district in Siem Reap Provinces in Cambodia indicated that using predicted $ET_c$ as the basis for managing irrigation applications was an appropriate approach, with 2–3-fold improvements in $WP_I$ compared to farmer practice.

However, the data gathering and computational processes of calculating $ET_c$ are not farmer-friendly in virtually any situation, including amongst the smallholder farmers represented in this study. Therefore, to facilitate effective changes to irrigation practice, it was necessary to develop simple tools that would enable farmers to estimate their crop water demand. The $ET_c$ approach has limitations (e.g., no account is taken of rainfall in the estimation of crop evapotranspiration), and will likely not provide optimal irrigation scheduling in extreme seasons when the management practices required will differ from most seasons [30]. Although there are more accurate methods of optimising irrigation schedules [31–33], it is emphasised that current irrigation practices widely used by smallholder farmers in Laos and Cambodia apply approximately twice the volume of water required by the crop. The tools outlined below may over- or under-estimate crop demand on any given day, but the authors contend that a potential water saving of over 50% compared to current practice, without yield penalty, is a clear advantage for a relatively simple method of irrigation scheduling and without any need to change application technology.

### 3.4.1. Daily Application Tables

The first step in the development of tools for use by farmers and advisors was the creation of daily application tables (Figure 4 and Supplementary Materials Files S1 and S2). These were developed for the dry season production period for each district or province involved in the broader program of work encompassing this study [19] (Xaithany, Hadxaifong, Paksong, and Phuntong in Laos, and Kampot, Siem Reap, and Phnom Penh in Cambodia), and for common leafy vegetable crops of different growing duration, namely short term (25–35 days from transplant to harvest), medium term (40–50 days), and long term (55–65 days). Examples of the commonly grown leafy vegetables in the study regions were short term—lettuce (*Lactuca sativa)*, mustard green (*Brassica juncea*), and Bok choy (*Brassica rapa* subsp. *chinensis*); medium term—kale (*Brassica oleracea* var. *acephala*), and long term—broccoli (*Brassica oleracea* var. *italica*) and cauliflower (*Brassica oleracea* var. *botrytis*). There was sufficient difference in predicted $ET_c$ at the district level in Laos to develop tables for each district, but this was not the case for the provinces of interest in Cambodia.

The tables were based on $ET_o$ calculated as the daily average from 33 years of NASA POWER Agroclimatology data for each of the districts and provinces involved. Crop demand ($ET_c$) was calculated from $ET_o$ and the relevant crop factor ($k_c$) based on crop similarity and crop growth stages. Drawing on information from Allen, Pereira, Raes and Smith [15] for broccoli, cabbage, cauliflower, lettuce and spinach, the following values of $k_c$ were used for all leafy vegetables–$k_{c\ (initial)}$ 0.7 and $k_{c\ (mid)}$ 1.05. As leafy vegetables are harvested fresh, the $k_{c\ (mid)}$ value was used through until harvest. The use of $k_{c\ (initial)}$ was delayed until after it was judged that transplant shock had been overcome, usually a period of 5–7 days. As part of the calculation process, crop growth stages [15] were adjusted in proportion to predicted crop duration.

The recommended daily application rates approximated what would be obtained if being guided strictly by the Penman–Monteith calculation process. The first approximation arises from the use of a 5-day moving average of predicted daily $ET_o$ to smooth out major fluctuations. The maximum difference between the daily predicted $ET_o$ and the 5-day moving average $ET_o$ for any given day in the dry season, the main period of interest, was $\pm7\%$. Since advising smallholder farmers using watering cans and hoses to vary irrigation volumes daily would be impractical, the predicted $ET_o$ changes were simplified to blocks of time, the duration of which was determined by the rate of change of $ET_o$. Data were analysed to determine on which dates during the growing period the absolute cumulative daily change in $ET_o$ exceeded 10% of the $ET_o$ at the start of a given time block. These dates were used as step change points for adoption of a new $ET_o$. Therefore, the $ET_o$ used for any given block of time was determined by the $ET_o$ at the change date. An example of the difference between predicted $ET_o$ (5-day moving average) and the $ET_o$ used to calculate $ET_c$ for the Paksong daily application tables is shown in Figure 5. The step-change $ET_o$ can lead to an under- or over-estimate of crop demand, depending on whether $ET_o$ is in a

rising or falling trend during the growth period of interest. The error was calculated to be <10% most of the time.

## Irrigation application tables – Lao PDR

**Paksong**

Irrigation schedule for quick growing (25–35 days) dry season (1 Oct–30 Apr) leafy vegetable crops – e.g. lettuce, mustard green, Chinese cabbage, Pak choy, Bok choy

| Paksong | Days after transplanting | | | |
|---|---|---|---|---|
| | 1–5 | 6–7 | 8–17 | Through to harvest |
| **Transplanting date** | Application per day (mm or L/m²) | | | |
| Between 1–Oct and 28–Nov | 3 | 2 | 3 | 3 |
| Between 29–Nov and 21–Jan | 4 | 3 | 3 | 4 |
| Between 22–Jan and 13–Feb | 4 | 3 | 4 | 5 |
| Between 14–Feb and 30–Apr | 4 | 3 | 4 | 4 |

Irrigation schedule for medium term (40–50 days) dry season (1 Oct–30 Apr) leafy vegetable crops – e.g. Chinese kale

| Paksong | Days after transplanting | | | |
|---|---|---|---|---|
| | 1–5 | 6–11 | 12–27 | Through to harvest |
| **Transplanting date** | Application per day (mm or L/m²) | | | |
| Between 1–Oct and 28–Nov | 3 | 2 | 3 | 4 |
| Between 29–Nov and 21–Jan | 4 | 3 | 3 | 4 |
| Between 22–Jan and 13–Feb | 4 | 3 | 4 | 5 |
| Between 14–Feb and 30–Apr | 5 | 3 | 4 | 5 |

Irrigation schedule for longer (55–65 days) dry season (1 Oct–30 Apr) leafy vegetable crops – e.g. broccoli, cauliflower

| Paksong | Days after transplanting | | | |
|---|---|---|---|---|
| | 1–5 | 6–15 | 16–34 | Through to harvest |
| **Transplanting date** | Application per day (mm or L/m²) | | | |
| Between 1–Oct and 28–Nov | 3 | 2 | 3 | 4 |
| Between 29–Nov and 21–Jan | 4 | 3 | 3 | 4 |
| Between 22–Jan and 13–Feb | 4 | 3 | 4 | 5 |
| Between 14–Feb and 30–Apr | 5 | 3 | 4 | 5 |

**Figure 4.** An example of daily application tables published for use by agricultural advisors and smallholder farmers (for Paksong district, Champasak Province, Laos) showing recommended daily application rates for leafy vegetables of three different growth durations.

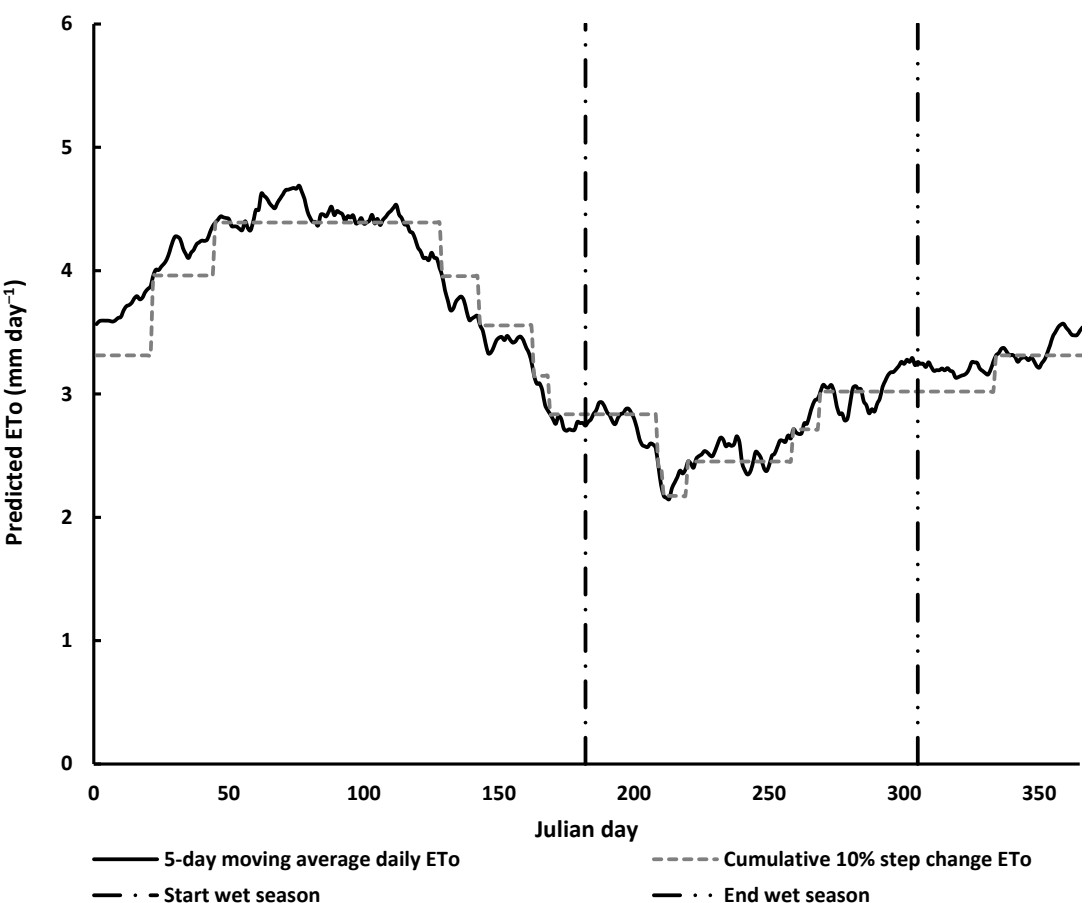

**Figure 5.** Five-day moving average $ET_o$, and approximate $ET_o$ based on absolute cumulative 10% change in $ET_o$ for Paksong district, Champasak Province, Laos.

Allowance was made to provide irrigation greater than plant demand in the days immediately after transplanting to allow the bare-rooted seedlings to overcome transplant shock. Application amounts were based on bare earth $ET_o$ for this period, after which the schedule followed the normal growth stage sequence outlined in Allen, Pereira, Raes and Smith [15].

Calculations of $ET_c$ were initially done to an accuracy of 0.1 mm (0.1 L/m$^2$), but such accuracy is unlikely to be obtained with the hand-held application technologies used by smallholder growers. Application rates 0.3 mm or more than one unit were rounded up to the next whole mm, while those less than 0.3 mm more than one unit were rounded down. Hence, the application rates in the tables (Figure 4) were always given in whole numbers.

3.4.2. Development of Nomograph

Provision of simplified daily application tables to guide growers in their irrigation management was the first step in improving irrigation management. However, few, if any, growers have the means to measure their application rate with any degree of accuracy or consistency. It was concluded that application advice had to be given in a form that was more meaningful to end users, such as the number of watering cans per crop bed, or the time required to water a crop bed knowing the output of a hand-held hose sprinkler. All these calculations are very simple to perform on a computer, calculator, or smart phone app. However, smart phones are not widely used in some areas, so while an app is a likely future step, an alternative approach was taken in the first instance.

A nomograph was developed (www.pynomo.org) (accessed on 4 June 2022) that allows advisors and farmers to calculate the number of watering cans or minutes of hose

watering based on knowledge of bed size (length and width in m), daily application rate (obtained from the appropriate table for district and crop duration), and system capacity, either watering can volume (L) or hose output (L/min). Figure 6 shows a worked example of the nomograph. A video explaining its use is available in Supplementary Materials. In this case, the daily application required to satisfy crop demand is 5 L/m$^2$ (5 mm), as for a 25–35 day crop that was transplanted between 22 January and 13 February in Paksong district, Laos and now approaching harvest (Figure 4). The crop bed is 10 m long by 1.2 m wide and a watering can of 12 litre capacity is used, giving the result that five watering cans are needed to provide the required amount. Likewise, if a hand-held hose with an output of 5 L/min was used, it would require five minutes of watering.

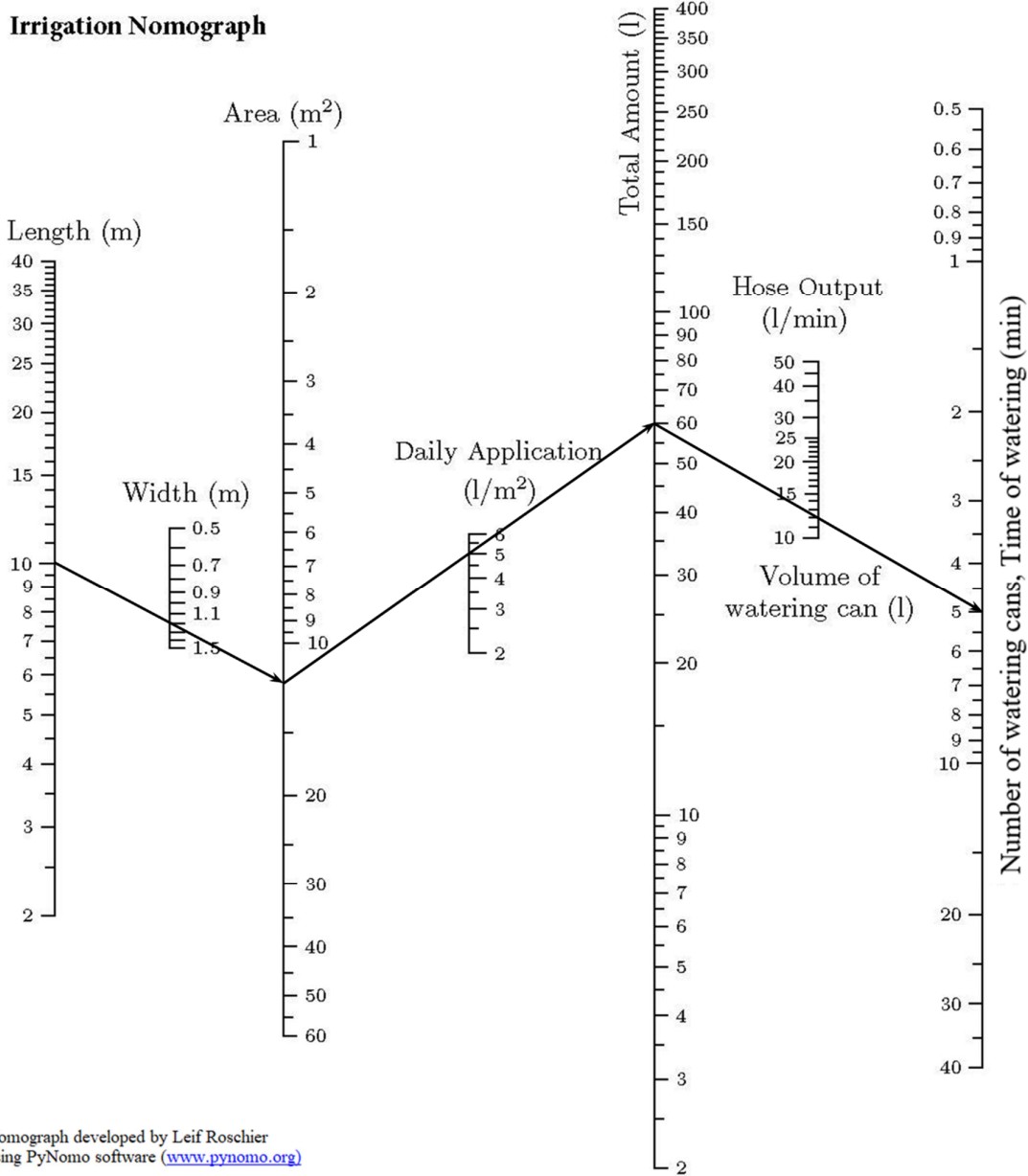

**Figure 6.** Nomograph developed for use by advisors and smallholder farmers in the study regions of Laos and Cambodia to calculate the number of watering cans, or duration of hose watering, required to apply the recommended daily irrigation application to leafy vegetables.

The tables, nomographs, and training materials (printed and video) developed as part of this work were produced in Lao and Khmer languages to aid adoption of the techniques by local agricultural advisors and farmers.

## 4. Discussion

### 4.1. Impact of Improved Irrigation Scheduling on Water Use and Yield

Given that the irrigation systems of interest in this study were watering cans and hand-held hoses, a change to $ET_c$-based watering, with total water application being approximately 50% of that previously used, would result in roughly equivalent savings of water, pumping costs (where relevant), and labour. The 50% reduction in water and commensurate reduction in labour use observed in this study compares well with the 43% water savings and 38% higher labour productivity made possible using low pressure drip irrigation technology in similar settings in Cambodia [5]. Prior research in Cambodia [3] found that low pressure drip irrigation technology required 3–9 times less labour than hand-watering, which was a bigger reduction than that achieved in this study through scheduling alone. Compared to hand-watering, low cost drip irrigation gave 15% higher yields for a range of vine and bush vegetable crops [5], while other research recorded yield increases ranging from 5–120% [3]. Although average yields of leafy vegetables increased with $ET_c$-based irrigation scheduling of hand-watering in this study (Table 5), the differences were not statistically significant.

### 4.2. Irrigation Scheduling Practice Change for Smallholder Vegetable Farmers

Reducing watering by 50% (i.e., change to once per day) was a practice readily adopted by some advisors, extension agents, and farmers after seeing the water and labour-saving advantages of $ET_c$-based irrigation scheduling [19]. This was a very easy change for farmers to make, once they were convinced there would be no yield penalty. However, despite widespread observation of the combined factors of over-application and twice a day watering, this is not necessarily current practice for all smallholder farmers. There is no guarantee that a particular crop will be over-watered by a factor of two, or even be over-watered at all, as was noted for some farms in Cambodia (Figure 3). While advice to water once a day is a very simple way to initiate change, such guidance ignores the nuances associated with differing water demand through the crop growth period and across the season, and could lead to adverse consequences (e.g., underwatering, yield loss, and/or loss of trust by farmers in the advice they receive). In some cases, extension and uptake of a 'water once a day' message preceded the development of the nomograph and associated training to assist with scheduling. While the change was enthusiastically adopted, there are risks associated with adoption in the absence of understanding the principles behind the method. The need for practice change in irrigation management was indicated in Ethiopian research [34]. This showed the combination of conservation agriculture and irrigation scheduling by estimated evapotranspiration could reduce the number of irrigations used on onions and garlic by 10–20%, the amount of water used by approximately 50% and improve irrigation water use efficiency by up to 80%, changes which are consistent with the work reported here.

### 4.3. Utility of $ET_o$-Based Irrigation Scheduling for Smallholder Vegetable Farmers and Their Advisors

While the provision of scheduling advice based on predicted $ET_c$ using the Penman–Monteith FAO 56 method [15] requires a considerable amount of data retrieval and processing, the outputs can be collated and presented in a form that is relatively easy to use by smallholder farmers. As part of this study, two tools were developed to assist with irrigation scheduling of smallholder leafy vegetable crops: 1. A series of daily application tables for locations and crops relevant to the study, and 2. A nomograph that uses information from the application tables and details of crop bed dimensions and irrigation system capacity to determine irrigation requirements in a form that is relevant to the operator of

the irrigation system. Extension personnel and farmers required less than half a day of training to use the tables and nomograph.

There is a significant amount of data retrieval and processing behind the development of the daily application tables and nomograph. The daily application tables are limited in that they are site specific and need to be developed for differing climatic conditions in different regions. Therefore, another process of data retrieval and processing is required for each location. On the other hand, the nomograph can be used universally as it draws on information from the daily application tables and details that are relevant to the specific farm.

The literature reports a wide range of research and development related to irrigation management using approaches based on estimated crop-water demand, with just a few examples being [20–22,24,35,36]. Some of the work done to date reports on the determination of crop water demand using predictive techniques, for example [21,22], while others report on improved mechanisms of prediction [20,37]. However, effective communication and adoption of these complex methods by farmers, particularly resource poor farmers, are rarely reported. Some have attempted to make the functionality of predictive techniques available through the use of spreadsheet and app-based calculators [26–28]. However, the novelty of this work is reflected in what appears to be an absence of literature outlining approaches to make the processes and outputs of crop water demand estimation methods, such as the Penman–Monteith method, available in a form that does not require complicated calculations and is therefore suited to smallholder farmers.

Other approaches to irrigation scheduling that have proven effective in other developing countries warrant further investigation for smallholder vegetable growers. These include low-cost evaporation pans made from locally available materials [38–42] and soil moisture sensing based on relatively cheap and simple sensors (e.g., Chameleon) [43]. One potential limitation of soil moisture sensing technologies is the range of different crops and planting dates that are a feature of leafy vegetable farming, such that there are likely to be numerous different irrigation requirements on a smallholder farm at any one time. Ideally, each of these scenarios would require their own set of sensors, although it is likely that with experience and participatory learning, farmers could manage a range of different crop growth scenarios with a limited set of sensors.

*4.4. Training and Future RD&E Needs*

Although the daily application tables and the nomograph are relatively simple tools, and were designed for ease of use, training is required to enable their use and application. Training was delivered via in-person workshops, and the production of written and audiovisual materials, primarily for the use of extension personnel and advisors (Supplementary Materials Video S1).

The daily application tables and nomograph are essentially printed copies of an 'app'. The next step in developing tools to assist advisors and farmers with irrigation scheduling would be the development of a smart phone app, which could be designed to improve the spatial and temporal accuracy of the estimated crop water demand. Such an app would embody all the background calculations associated with the Penman–Monteith method [15] that were performed using a Microsoft Excel® workbook, and allow the farmer to enter site specific information such as hose output, watering can volume, and crop bed dimensions.

An app could also be designed to retrieve historical climate data relevant to the users' geographical location, although there is always risk in over-complicating the process, particularly when web sites from which data are sourced change formatting or access requirements. Smart phone apps are not maintenance free, so ensuring usefulness in the long-term requires investment in maintenance and updating. Further, a mobile phone app developed for use in Bangladesh, while useful for researchers, proved less useful for farmers due to its perceived complexity [44]. This highlights the importance of developing understanding of, and training in, the principles of irrigation management alongside the

technology, while keeping the enabling technology as simple and relevant as possible for the target end users.

## 5. Conclusions

We condensed complex data and calculations from the Penman–Monteith FAO 56 method for calculating $ET_o$ into two simple tools to assist with irrigation scheduling of smallholder farmer leafy vegetable crops. These were location-specific daily application tables for crops of varying duration, and a nomograph. The nomograph uses information from the application tables and details of crop bed dimensions and irrigation system capacity to convert daily crop requirements into site specific guidance (e.g., number of watering cans, or duration of hose watering, required). When used together to inform field irrigation management, this novel approach of simplification achieved up to 50% savings in water, labour, and pumping costs.

Extension personnel and farmers required less than half a day of training to use the tables and nomograph. Using the tools and guidelines developed in this study has the potential to increase the volume and sustainability of leafy vegetable production amongst smallholder farmers in Laos and Cambodia, and so improve their livelihoods, particularly in a changing climate. The method is applicable in any region where smallholder farmers use irrigation in their production system. Through further development, a smart phone app could be useful to deliver irrigation advice in the future and improve the spatial and temporal accuracy of the estimated crop water demand.

**Supplementary Materials:** The following supporting information can be downloaded at: https://www.mdpi.com/article/10.3390/w14132010/s1, Table S1—Weather data for Phnom Penh, Cambodia; Table S2—statistical measures of weather data for Phnom Penh, Cambodia; File S1—Daily application tables for irrigation of leafy vegetables in Laos; File S2—Daily application tables for irrigation of leafy vegetables in Cambodia; Video S1—Irrigation application for leafy vegetables.

**Author Contributions:** Project conceptualization, development and oversight, S.I., J.M., J.E., A.M., S.H. and P.M.; experimental methodology design, field site management, data collection, analysis and interpretation, development of scheduling tools, J.M., J.E., J.U., A.M., S.H., V.T., V.L., P.S., I.S., P.M. and T.V.; writing—original draft preparation, J.M., L.D. and A.M.; writing—review and editing, J.M., A.M., J.U., J.E., S.H. and S.I. project administration, A.M. and S.I.; funding acquisition, S.I. and J.M. All authors have read and agreed to the published version of the manuscript.

**Funding:** This work was funded by the Australian Centre for International Agricultural Research (ACIAR), Canberra, ACT, Australia (grant number SMCN/2014/088—Integrating soil and water management in vegetable production in Laos and Cambodia).

**Acknowledgments:** Thanks is given to Bounneuang Douang Boupha and Vang Seng who were, respectively, Horticultural Research Institute, Vientiane, Laos and Deputy, Cambodian Agricultural Research and Development Institute, Phnom Penh, Cambodia during the conduct of the project. Suzie Jones, University of Tasmania, is acknowledged for her contribution at various stages of the project and for review of the draft manuscript. Ann Starasts and Gomathy Palaniappan are acknowledged for their contribution to extension material development, partner training and extension facilitation that promoted the use of the irrigation scheduling methods developed. Thanks is given to Leif Roschier (www.pynom.org, accessed on 4 June 2022) for development of the nomograph and to Ross Corkrey for statistical analyses.

**Conflicts of Interest:** The authors declare no conflict of interest. The funders had no role in the design of the study; in the collection, analyses, or interpretation of data; in the writing of the manuscript, or in the decision to publish the results.

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
