# Peer review of "Simple ETo-Based Rules for Irrigation Scheduling by Smallholder Vegetable Farmers in Laos and Cambodia"

_water, doi:10.3390/w14132010_

Round 1

Reviewer 1 Report

Thank you for this paper. It will help the farmers to reduce work and the world to save water.

Author Response

No response required. Suggestions regarding improvements have been addressed through the responses to Reviewers 2 and 3.

Reviewer 2 Report

The abstract needs to be restructured. It should have following order:

  • Background: Place the question addressed in a broad context and highlight the purpose of the study;
  • Methods: briefly describe the main methods applied;
  • Results: summarize the article’s main findings;
  • Conclusions: indicate the main conclusions or interpretations.

Introduction section – Some of the references are very outdated. Please add more recent references and ensure the paper make adequate reference to earlier articles published in the Water MDPI journal. Please cite ‘Enhancement of a Spent Irrigation Water Recycling Process: A Case Study in a Food Business’. Also, there are quite a few paragraphs in this section. I recommend the authors to break it down into five:

  • Briefly place the study in a broad context and highlight why it is important.
  • Define the purpose of the work and its significance.
  • The current state of the research field should be carefully reviewed and key publications cited. One of the area that is being highly pursued is the application of IoT, Big Data, AI etc. Please see and cite following articles ‘Improving Water Efficiency in the Beverage Industry with the Internet of Things’ ‘An Internet of Things Approach for Water Efficiency: A Case Study of the Beverage Factory’  
  • Please highlight controversial and diverging hypotheses when necessary.
  • Finally, briefly mention the main aim of the work and highlight the principal conclusions. Do not forget to define Penman-Monteith method.

Materials & Methods, Results, Discussion and Conclusion section flows well.

Reviewer 3 Report

The publication entitled ,,Simple ETo-Based Rules for Irrigation Scheduling by Small-2 Holder Vegetable Farmers in Lao PDR and Cambodia” is a significant source of knowledge on irrigation for local farmers. The work has a practical aspect which is very important. However, when considering the paper as a scientific work, it is unclear what is new and what the analyses carried out actually contribute to the current state of knowledge. This should be better highlighted. The methods used by the authors have been known for years and I do not see the novelty of the paper. Moreover, the following corrections should be made in the manuscript:

1) The authors should state the value of the kc coefficient used in the calculations and its source. The choice of this coefficient has a significant impact on the results obtained

2) The discussion section should be expanded. The results obtained by the authors should be more widely compared with previously published results.

3) The conclusions section lacks a clear statement of what new findings the authors have established for the state of knowledge, which would be worth publishing and of interest to international readers. Their analyses are valuable but, in my opinion, only of local relevance for farmers. This chapter should be rewritten.

4) The abstract lacks the aim of the research.

Round 2

Reviewer 3 Report

I appreciate the authors' contribution to improving the manuscript. Thank you for all your responses to my comments. I believe the article is suitable for publication. The only thing I might have a comment on is that the authors refer to relatively few literature items in the references, and I still think the discussion could be developed further. I would ask the authors to check the English carefully before publishing and edit the list of references to the journal's requirements.

Author Response

The authors refer to relatively few literature items in the references - the authors consider that the relevance of the cited references is more important than the number of cited references. All of the references cited in the paper relate to the statements made and we don't consider it is necessary to overload the paper with more references for no obvious benefit.

The discussion could be developed further - no insight has been given as to what is missing from the discussion, so it is difficult for the authors to second guess what the reviewer considers is missing. The authors stand by the discussion as written.

Check the English carefully before publishing - some minor punctuation oversights have been corrected.

Edit the list of references to the journal's requirements - the reference list has been compiled in EndNote using the Water style.
